# Evaluating the Use of Sacran, a Polysaccharide Isolated from *Aphanothece sacrum*, as a Possible Microbicide for Preventing HIV-1 Infection

**DOI:** 10.3390/v16091501

**Published:** 2024-09-23

**Authors:** Kouki Matsuda, Ryusho Kariya, Kenji Maeda, Seiji Okada

**Affiliations:** 1Division of Hematopoiesis, Joint Research Center for Human Retrovirus Infection, Kumamoto University, 2-2-1 Honjo, Chuo-ku, Kumamoto 860-0811, Japan; kmatsuda@kufm.kagoshima-u.ac.jp (K.M.); ryusho@kumamoto-u.ac.jp (R.K.); 2Division of Antiviral Therapy, Joint Research Center for Human Retrovirus Infection, Kagoshima University, 8-35-1, Sakuragaoka, Kagoshima 890-8544, Japan; kmaeda@kufm.kagoshima-u.ac.jp

**Keywords:** sacran, HIV-1, microbicide

## Abstract

Since combination antiretroviral therapy (cART) was introduced to treat human immunodeficiency virus type-1 (HIV-1)/acquired immunodeficiency syndrome (AIDS), the AIDS mortality rate has markedly decreased, and convalescence in individuals with HIV has improved drastically. However, sexual transmission has made HIV-1 a global epidemic. Sacran is a megamolecular polysaccharide extracted from cyanobacterium *Aphanothece sacrum* that exhibits numerous desirable characteristics for transdermic applications, such as safety as a biomaterial, a high moisture retention effect, the ability to form a film and hydrogel, and an anti-inflammatory effect. In this study, we evaluated the anti-HIV-1 effects in sacran as a barrier to HIV-1 transmission. Sacran inhibited HIV-1 infection and envelope-dependent cell-to-cell fusion. Moreover, we used a Transwell assay to confirm that sacran inhibited viral diffusion and captured viruses. The synergistic effects of sacran and other anti-HIV infection drugs were also evaluated. HIV-1 infections can be reduced through the synergistic effects of sacran and anti-HIV-1 drugs. Our study suggests using sacran gel to provide protection against HIV-1 transmission.

## 1. Introduction

Over the past decade, the human immunodeficiency virus type-1 (HIV-1)/acquired immunodeficiency syndrome (AIDS) epidemic has led to the development of several HIV-1 testing technologies to prevent its spread [1]. Novel prevention methods that combine advances in drug development and delivery can contribute significantly to slowing down the incidence of HIV-1 infections. Most human HIV-1 infections are sexually transmitted through infected semen and vaginal or cervical secretions containing infected lymphocytes [2]. According to a 2017 study, the number of newly infected adolescent girls and young women (AGYW) aged 15–24 is very high, with infection rates reaching up to 1000 per day. In sub-Saharan Africa, in particular, AGYW are twice as likely to be infected with HIV-1 than men; 75% of newly reported infections among adolescents aged 15–19 are detected in girls [3]. Male condoms are the most widely publicized HIV-1 prevention measure, but this method relies on men, and this may limit the ability of more vulnerable women to manage their own HIV-1 prevention. Pre-exposure pro-phylaxis (PrEP) is a very effective means of HIV-1 prevention, but it requires strict adherence to daily medication regimens, which have been shown to fail in some cases. The emergence of drug-resistant viruses with PrEP has also been reported [4], but adherence problems have been pointed out as a possible cause, and it can be difficult to manage a fixed oral dosing schedule.

Microbicides can be applied to the vagina and rectum to reduce the transmission of sexually transmitted infections (STIs), including HIV-1 [5]. Vaginal microbicides are a potential means of protecting women against HIV-1 infection during sexual transmission [5]. Thus, a microbicide is a potential female-led biomedical intervention for HIV-1 prevention when the female cannot negotiate condom use. However, there is still a high prevalence of HIV among men who have sex with men (MSM), and frequent anal intercourse is a risk factor. Therefore, microbicides may contribute to the HIV-1 infection risk reduction not only for women but also for men. Early-generation microbicide candidates based on detergents and polyanions have failed to achieve protective effects in efficacy tests, and some products even promote viral transmission [6]. Although tenofovir (TFV)-based gels have been shown to decrease HIV-1 infection acquisition [7], the FACTS 001 clinical trial showed a failure of protection [8].

The ampholytic polysaccharide sacran was extracted from cyanobacterium *Aphanothece sacrum*, a species indigenous to Japan that is massively cultured in rivers with high ionic concentrations, and it is rich in a high-water-content (97.5%) jelly-like extracellular matrix [9,10,11,12,13]. Sacran is a heteropolysaccharide that contains a variety of sugar residues, including the following: glucose, galactose, mannose, xylose, rhamnose, fucose, galacturonic acid, glucuronic acid, and traces of alanine, galactosamine, and muramic acid [14]. In addition, 11% of the monosaccharides contain sulfate groups, and 22% contain carboxyl groups and have an extremely high molecular weight (approximately 20 MDa) and are surprisingly long (over 8 μm) [9]. Sacran is produced from cyanobacteria and is considered to be a safe biomaterial because it has traditionally been used as a functional food to improve allergic tendencies and gastroenteritis in the Kyushu region of Japan [14]. In addition, sacran can hold a large amount of water compared with hyaluronic acid or xanthan gum and can form a nanofilm on the platform [13]. Sacran is highly biocompatible because it is a polysaccharide derived from edible cyanobacteria and is used in a variety of states, including viscous solutions, heterogels, and gel beads. These gel-forming and film-forming abilities are particularly unique, and heterogels conditioned with polyvinyl alcohol (PVA) have a high degree of swelling and improve the water absorption of sacran [11]. Sacran also has excellent metal adsorption, more efficiently adsorbing heavy metal ions such as indium, a rare earth metal, and leading ions to form gel beads [10].

In this study, we investigated the effect of a viscous sacran gel as an anti-HIV-1 agent using a Transwell assay that mimics HIV-1 transmission by macrophages/dendritic cells and T lymphocytes in mucosal tissue. Our findings suggest a potential application of sacran in preventing HIV-1 transmission.

## 2. Materials and Methods

### 2.1. Preparation of Sacran Gel

Sacran gel was kindly provided by Green Science Materials (Kumamoto, Japan), and analytical reagent-grade solvents were used for all experiments [10,11,12]. The sacran gel was diluted with deionized double-distilled water.

### 2.2. Cell Lines and Culture

The TZM-bl, Jurkat_HXBc2 (4)_, and Jurkat_522F/Y_ cells were procured from the National Institutes of Health AIDS Reagent Program (Bethesda, MD, USA). The MOLT-4 T cell line was obtained from the RIKEN cell bank (Tsukuba, Japan). The MT-4 T cell line was kindly provided by Dr. Kazuhiko Ide (Kumamoto University, Kumamoto, Japan). The human T cell line PM1-CCR5, stably expressing human CCR5 [15], was kindly gifted by Dr. Yosuke Maeda (Kumamoto University). The MT-4, MOLT-4, and PM1-CCR5 cells were maintained in an RPMI 1640 medium (Fujifilm Wako Pure Chemical, Osaka, Japan) supplemented with penicillin 100 U/mL, streptomycin 100 μg/mL, and 10% fetal bovine serum (Thermo Fisher Scientific, Waltham, MA, USA) and cultured in a 5% CO_2_ humidified incubator at 37 °C. Stably expressing HIV-1 env cell lines, Jurkat_HXBc2 (4)_, and Jurkat_522F/Y_, were used for the cell–cell fusion assay and cultured in RPMI 1640 supplemented with 10% FBS, 100 U/mL penicillin, 100 µg/mL streptomycin, 1 µg/mL doxycycline, 200 µg/mL G418, and 200 µg/mL hygromycin. Jurkat_HXBc2 (4)_ expresses functional gp120 and gp41 glycoproteins (*Env*), whereas Jurkat_522F/Y_ has an F/Y substitution at position 522 of gp41, which prevents fusion [16]. HIV-1 env expression was induced by a Tet-Off system, whereby cells were washed with PBS and cultured in a doxycycline-free medium for 3 days prior to the fusion experiments [17,18]. The TZM-bl were maintained in DMEM (Fujifilm Wako Pure Chemical) supplemented with penicillin 100 U/mL, streptomycin 100 μg/mL, and 10% fetal bovine serum.

### 2.3. Virus Preparation

HIV-1_LAI/IIIB_ was procured from the NIH AIDS Reagent Program. The viruses were propagated in the MT-4 HVLV-1-infected T cell line, and the supernatants were collected after 4 days. The amount of virus was quantified with a p24 antigen ELISA kit (ZeptoMetrix Corp., Buffalo, NY, USA) and stored at −80 °C until use.

### 2.4. Flow Cytometric Detection of HIV-1-Infected Cells

The viral infections were performed under three experimental settings. First, PM1-CCR5 T cell line (5 × 10^5^ cells/mL), sacran gel (final concentration 0–0.2%), and HIV-1LAI (X4 tropic) (p24 concentration: 25 ng/mL) were cultured in a triple-layered medium for 48 h at 37 °C (Figure 1A, left (a)). Second, sacran gel (final concentration 0–0.2%) and HIV-1LAI (p24 concentration: 25 ng/mL) were co-treated with PM1-CCR5 cells and cultured for 48 h at 37 °C (Figure 1A, center (b)). Third, sacran gel (final concentration 0–0.2%) and HIV-1LAI (p24 concentration: 25 ng/mL) were mixed in a 1.5 mL tube. Then, the mixed solution was treated with PM1-CCR5 cells and cultured for 48 h at 37 °C (Figure 1A, right (c)). Intracellular p24-positive cells were identified as HIV-1-infected using flow cytometric analysis. Briefly, cells were fixed with 1% paraformaldehyde in PBS for 20 min in the dark and permeabilized with 0.1% saponin in PBS. Following incubation on ice for 10 min, the cells were stained with anti-HIV-1 Gag p24-FITC mAb (Beckman Coulter, Fullerton, CA, USA) on ice for 30 min [17,19]. The stained cells were analyzed using an LSR II flow cytometer (BD Biosciences, Franklin Lakes, NJ, USA). All corrected data were analyzed using the FlowJo software version 9 (Tree Star, San Carlos, CA, USA).

### 2.5. Cytotoxicity Assay

To examine the cytotoxicity of sacran gel, several lineages of human T cell lines, PM1-CCR5, MOLT-4, and Jurkat cells (5 × 10^5^ cells/mL) were cultured in the presence or absence of sacran gel (final concentration 0–1%). After 24 h, a cytotoxicity assay (WST-8 assay) was performed using the Cell Counting Kit-8 (Dojindo, Kumamoto, Japan) according to the manufacturer’s instructions. The number of viable cells after the sacran gel treatment was measured and compared with the untreated cells.

### 2.6. Cell–Cell Fusion Assay

The effector cells, the HIV-1 env-expressing cell lines (Jurkat_HXBc2 (4)_, Jurkat_522F/Y_), and the target cells, the MOLT-4 T cell lines, were labeled using the PKH 67 Green Fluorescent Cell Linker Kit (Sigma-Aldrich, St. Louis, MO, USA) and the PKH 26 Red Fluorescent Cell Linker Kit (Sigma-Aldrich), respectively. These two cell populations were then cocultured at a ratio of 1:1 for 24 h and treated with sacran gel at a final concentration of 0–0.2%. The cells were analyzed using an LSR II flow cytometer, and double-positive cells for PKH67 and PKH26 were defined as fusion cells. Data were analyzed using FlowJo software.

### 2.7. Transwell Experiments to Test the Barrier Function of Sacran

We applied 50 µL of sacran gel to the membranes of the 24-well Transwell^®^ plates (pore size = 8 µm; Corning Inc., Corning, NY, USA). Each gel layer was challenged with a solution of 50 µL of HIV-1 virions (p24 concentration: 2 µg/mL) added to the upper chamber. The bottom compartment of each Transwell contained 600 µL of 10% FCS/DMEM. The Transwell plates were incubated in a 5% CO_2_ incubator at 37 °C for 24 h. Following incubation, the Transwell plates were removed from the incubator, and the solutions were collected from the upper (sacran gel-containing) and lower chambers. The amount of p24 was measured in samples from both the upper and lower chambers using a p24 antigen ELISA kit.

### 2.8. Transwell Experiments for TZM-bl Assay

Three different solutions (37.5 µL each) were applied to the membranes of the 96-well Transwell^®^ plates (pore size = 5 µm; Corning Inc.): pure sacran gel, sacran gel mixed with an HIV-1 entry inhibitor (either plerixafor/AMD3100 [20] or enfuvirtide/T-20 [21]), and sacran gel mixed with the HIV-1 reverse transcriptase inhibitor efavirenz (EFV) [22]. Each gel layer was challenged with a solution of 37.5 µL of HIV-1 virions (p24 concentration: 533 ng/mL) added to the upper chamber. The bottom compartment of each Transwell contained 235 µL of 10% FCS/DMEM. Transwell plates were incubated in a 5% CO_2_ incubator at 37 °C for 24 h. After incubation, the solutions were collected from the bottom chambers and used for the TZM-bl assay.

### 2.9. TZM-bl Assay

The TZM-bl assay was used to assess the synergistic effects of sacran and HIV-1 entry inhibitors were used [23,24]. TZM-bl cells were used as viral target cells, seeded in the 96-well tissue culture plates at a density of 1 × 10^4^ cells/well, and cultured for 24 h. The HIV-1 solution collected from the bottom chambers (as described in the Materials and Methods in Section 2.8) was used to infect the cells. The infectivity of viruses was evaluated by induction of HIV-1 Tat-mediated β-galactosidase activity in the target cells at 48 h post-infection using a mammalian β-galactosidase assay kit (Thermo Fisher Scientific Inc., Waltham, MA, USA) [24]. The absorbance of the wells was measured at 405/595 nm using an iMark microplate absorbance reader (Bio-Rad, Hercules, CA, USA).

### 2.10. Quantification of Synergistic Effects of Sacran with Anti-HIV-1 Agents

The effects of sacran in combination with HIV-1 entry inhibitors or HIV-1 reverse transcriptase inhibitors on HIV-1 infection were calculated with the Chou and Talalay method utilizing the CompuSyn software (ComboSyn, Inc., Paramus, NJ, USA) based on the quantitative analysis of the concentration–effect relationships of multiple drugs or enzyme inhibitors by Chou and Talalay [25,26,27]. The software calculates a combination index (CI) value to confirm the synergistic effects and compares it to the effect of a single agent to further help determine the properties of the combination. CI values < 1 indicate synergistic effects, CI values equal to 1 indicate a mean additive effect of the drugs, and CI values > 1 indicate antagonistic effects.

### 2.11. Statistical Analysis

Statistical parametric analysis was conducted using the unpaired *t*-test. The *p*-values set at >0.05 were considered statistically significant.

## 3. Results

### 3.1. Sacran Inhibited HIV-1 Infection

We tested the response of an HIV-1 infection to sacran gel treatment on the PM1-CCR5 cells [15] in three experimental settings. A human CD4 and human CXCR4/CCR5-positive T cell line, PM1-CCR5 cells, were exposed to the X4 tropic virus HIV-1LAI/IIIB with or without sacran gel. First, the target cells, sacran gel, and HIV-1 were cultured in a triple-layered medium for 48 h at 37 °C (Figure 1A(a) and Figure 1B(a)). The intracellular expression of p24 decreased after this sacran gel treatment in a concentration-dependent manner (Figure 1C). Next, the target cells were treated with sacran gel and HIV-1 and cultured for 48 h at 37 °C (Figure 1A(b) and Figure 1B(b)). The intracellular expression of p24 was decreased by the sacran gel treatment in a concentration-dependent manner (Figure 1D). Third, sacran gel and HIV-1 were mixed in a 1.5 mL tube. The target cells were then treated with sacran and the HIV-1 mix solution for 48 h at 37 °C (Figure 1A(c) and Figure 1B(c)). The intracellular p24 expression was decreased by this sacran gel treatment in a concentration-dependent manner (Figure 1E). These results indicated that sacran gel inhibited HIV-1 infection. The cytotoxicity of sacran gel was also examined using several human T cell lines, PM1-CCR5, MOLT-4, and Jurkat cells. Sacran gel had no toxicity in these cells at concentrations up to 1% (see Appendix A).

### 3.2. Sacran Inhibited HIV-1 Envelope-Dependent Cell Fusion

The membrane fusion between the virus-infected and target cells is an essential step in HIV-1 infection. Therefore, we used an in vitro fusion model system to evaluate the effect of sacran gel on HIV-1 envelope-dependent cell fusion [16,18]. PKH 67 (green fluorescence)-labeled env-expressing cells, Jurkat_HXBc2 (4)_ (effector cells), and PKH 26 (red fluorescence)-labeled MOLT-4 T cells (target cells) were cocultured at a ratio of 1:1 for 24 h with or without sacran gel at a final concentration of 0–0.2%. The Jurkat_522F/Y_ cells were used for a fusion negative control. As shown in Figure 2A, cells that were double-positive for PKH 67 and PKH 26 were defined as fused cells, which decreased in a concentration-dependent manner following the sacran gel treatment (Figure 2A). Sacran gel significantly inhibited cell fusion (*p* < 0.001) (Figure 2B). These results demonstrate that sacran gel inhibits HIV-1 envelope-dependent cell–cell fusion. This suggests that sacran gel might provide protection against cell–virus interactions.

### 3.3. Sacran Inhibited Viral Diffusion and Captured HIV-1

Transwell assays were used to assess the impact of viral diffusion and the ability of sacran to capture the virus [23]. Sacran gel was applied to the membranes of the 24-well Transwell plates. Each gel layer was challenged with HIV-1 virions and added to the upper chamber. Following incubation, the solutions from the upper (sacran gel containing) and lower chambers were collected, and the p24 amounts of the samples were measured using a p24 antigen ELISA kit (Figure 3A). As shown in Figure 3B, following the sacran treatment, p24 levels in the lower chamber decreased in a concentration-dependent manner. The p24 levels in the upper chamber increased following the sacran treatment in a concentration-dependent manner (Figure 3C). A marked inhibition of viral diffusion has been observed at high concentrations (0.5–1%) of sacran. This may be related to the viscosity of sacran and may involve the physical properties of sacran, which is a macromolecule. These results demonstrate that sacran gel inhibits HIV-1 diffusion and has a viral capture ability.

### 3.4. Synergistic HIV-1 Inhibition of Sacran and Anti-HIV-1 Drugs

Sacran gel or HIV-1 entry inhibitors, plerixafor or enfuvirtide-containing sacran gel, were applied to the membranes of the 96-well Transwell plates. Each gel layer was challenged with HIV-1 virions and added to the upper chambers. Following incubation, the solutions from the bottom chambers were collected and used for the TZM-bl assay. As shown in Figure 4A,B, the HIV-1 inhibitory activities of sacran with HIV-1 entry inhibitors were significantly enhanced in a concentration-dependent manner (Figure 4A). The combination index was calculated using the CompuSyn software [18] according to the method previously established by Chou and Talalay [19,20] to evaluate the synergistic effect of sacran and HIV-1 entry inhibitors. The CI values for sacran and HIV-1 entry inhibitors were <1 (Table 1). This indicated that sacran exhibited synergism with plerixafor and enfuvirtide (Figure 4B,C). Moreover, we evaluated the synergistic effect of sacran with the HIV-1 reverse transcriptase inhibitor, efavirenz (EFV), using the TZM-bl assay. Sacran gel or EFV-containing sacran gel was applied to the membranes of the 96-well Transwell plates. The HIV-1-inhibiting activities of sacran with EFV were significantly enhanced in a concentration-dependent manner (Figure 5A). The CI values for the sacran and EFV were <1 (Table 1). This indicates that sacran acts synergistically with EFV (Figure 5B). These results suggest synergism between sacran and anti-HIV-1 drugs.

## 4. Discussion

Sexual transmission is a major contributor to the HIV-1 infection pandemic. The prevention of sexual transmission is beneficial for HIV-1 eradication. The development of tools to prevent viral transmission has shown that the TFV-based gel formulations currently prescribed in PrEP reduce the chance of HIV-1 transmission. The CAPRISA 004 trial was the first large trial to demonstrate the efficacy of 1% TFV gel, which reduced 39% of heterosexual HIV-1 infections [7]. However, the FACTS 001 clinical trial, in which 1% TFV gel was prescribed within 12 h before and after sexual intercourse, showed no preventive effects in women [8]. This study was evaluated based on the number of empty gel applicators returned and self-reported sexual activity; the lack of efficacy may be attributed to low adherence. The HIV-1 entry inhibitors enfuvirtide and maraviroc, which are used in clinical HIV-1-infected individuals, suppress the HIV-1 fusion with its target cells and prevent HIV-1 infection by specifically inhibiting the virus–receptor interactions or subsequent steps in the entry process [28,29,30]. Several entry inhibitors have been shown to prevent SHIV infection in macaques through vaginal and/or rectal transmission [31]. Notably, the gp120-binding inhibitor BMS-378806, gp41-targeted fusion inhibitor peptide C52L, and small-molecule CCR5 inhibitor CMPD167 were significantly protected by each compound alone and in combination in vivo. Additionally, the combination of the small-molecule CXCR4 inhibitors AMD3465 and CMPD167 inhibits R5X4 HIV-1 [32]. Therefore, the combination of entry inhibitors may improve the efficacy of microbicides. In this study, we demonstrated a combination assay of sacran with the small-molecule CXCR4 inhibitor plerixafor, gp41-targeted fusion inhibitory peptide enfuvirtide, and reverse transcriptase inhibitor efavirenz. The sacran gel alone was effective against HIV-1 infection, but its combination with the anti-HIV-1 drugs strongly inhibited HIV-1 infection due to the synergistic effects (Figure 4 and Figure 5). These results suggest the potent effects of sacran in combination with anti-HIV-1 drugs.

Sacran is a sulfated polysaccharide isolated from the cyanobacterium *Aphanothece sacrum* and has the potential for biomass applications [10]. In this study, we investigated the anti-HIV-1 activity of sacran gel and found that it suppressed HIV-1 infectivity (Figure 1). Cell-to-cell transmission plays a critical role in the spread of HIV-1 [33]. The encounter of the HIV-1 envelope with the target cells causes transmission and influences its efficacy. As demonstrated in this study, the sacran gel treatment inhibited HIV-1 infection derived from cell-to-cell fusion, which mimicked macrophage/dendritic cell and T lymphocyte transmission (Figure 2).

Compounds extracted from marine and freshwater micro/macroalgae have shown antiviral potency against a variety of retroviruses (HIV-1/SIV), herpes viruses (HSV-1, HSV-2, and HCMV), rhabdoviruses (VSV), and paramyxoviruses (RSV) [34]. Most studies have evaluated the anti-HIV effects of natural and synthetic sulfated polysaccharides because the antiviral activity is affected by the degree of sulfation and a high molecular weight. In addition, some compounds isolated from algae selectively inhibit the replication of enveloped viruses, such as HSV-2, HCMV, RSV, influenza A and B viruses, and SIV [35]. Their modes of action against these viruses include an inhibitory effect on viral adsorption. We performed a Transwell assay to confirm the antiviral effect of the functional material. Following the sacran gel treatment, the p24 levels in the lower chamber decreased, whereas those in the upper chamber increased in a concentration-dependent manner (Figure 3). These results suggest that sacran has an antiviral diffusion ability and an inhibitory effect on viral adsorption.

Sacran has already been applied in cosmetic products, such as skin toners and face creams, because of its water retention capacity, moisturizing effect, and film-forming ability. Furthermore, it has been reported to be an effective treatment for induced allergic dermatitis in mice. In addition, sacran significantly reduced transepidermal water loss in individuals with dry skin [36]. Notably, sacran controlled the itchiness and blood IgE concentrations.

Our findings demonstrate a potential anti-HIV-1 effect of sacran and a synergistic effect of sacran and anti-HIV-1 agents in in vitro models. However, the efficacy of sacran gel against HIV-1 infection in vivo requires further investigation to facilitate the development of an effective microbicide.

## 5. Conclusions

We confirmed that sacran inhibited HIV-1 infection by reducing viral diffusion and capturing viral particles. The combination of sacran and anti-HIV-1 agents significantly reduces HIV-1 infection via synergistic effects. Our study suggests that sacran gel could potentially be used for protection against HIV-1 transmission.

## Figures and Tables

**Figure 1 viruses-16-01501-f001:**
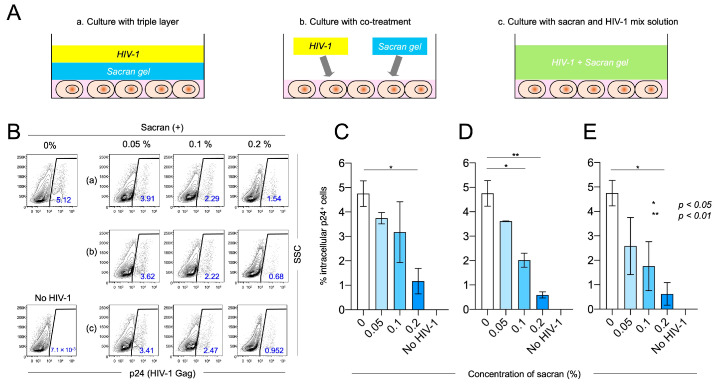
HIV-1_LAI_-infected PM1-CCR5 cells were analyzed using flow cytometry 48 h post-infection. (**A**) HIV-1 infection was induced in all three experimental settings. (a) Target cells, sacran gel, and HIV-1 were cultured with a triple-layered gel for 48 h at 37 °C. (b) Sacran gel and HIV-1 were co-treated to target cells and cultured for 48 h at 37 °C. (c) Sacran gel and HIV-1 were mixed in a 1.5 mL tube. Then, sacran and HIV-1 mix solution were added to target cells and cultured for 48 h at 37 °C. (**B**) Intracellular p24 (HIV-1 Gag) levels in sacran-treated PM1-CCR5 cells. (**C**–**E**) Summary of intracellular p24 levels in three experimental settings (a–c), respectively. Data represent the mean ± SD. Representative results from three independent experiments are shown. * *p* < 0.05, ** *p* < 0.01.

**Figure 2 viruses-16-01501-f002:**
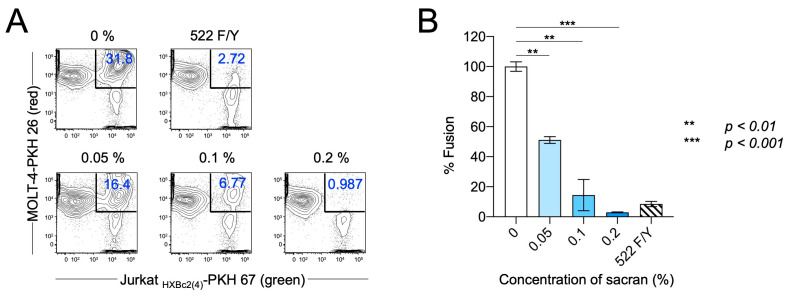
Sacran inhibited env-dependent cell-to-cell fusion. (**A**) Inhibition of cell fusion was analyzed using flow cytometry. (**B**) Summary of (**A**). Data represent the mean ± SD. One representative result from three independent experiments is shown.

**Figure 3 viruses-16-01501-f003:**
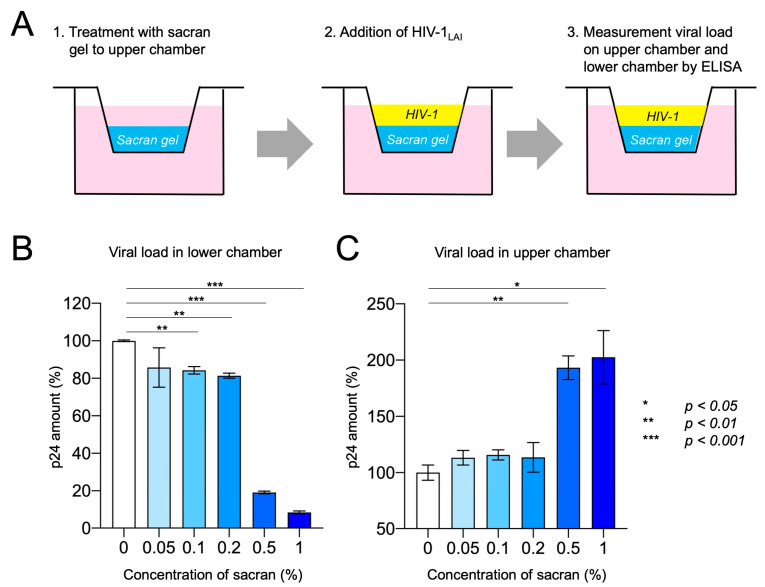
Sacran inhibited viral diffusion and captured HIV-1. (**A**) Experimental design. The Transwell system simulates HIV-1 transmission in the presence of a vaginal gel. A thin gel layer was applied to the Transwell membrane. A suspension of HIV-1 was then added to the upper compartment. After incubation, the levels of HIV-1 in the upper and lower chambers were quantified using p24 ELISA. (**B**) Viral loads in the lower chamber. (**C**) Viral load in the upper chamber. Data represent the mean ± SD. Representative results from three independent experiments are shown.

**Figure 4 viruses-16-01501-f004:**
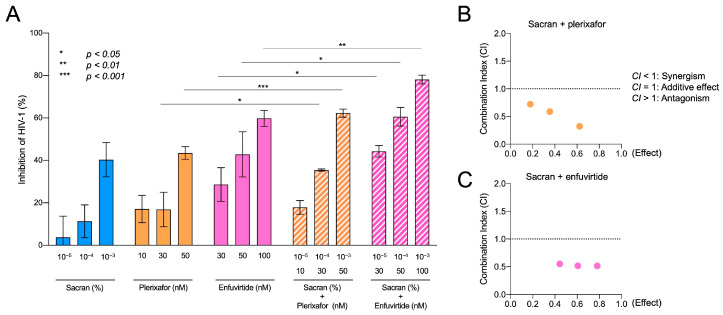
Synergistic HIV-1 inhibition of sacran and HIV-1 entry inhibitors. (**A**) HIV-1 inhibitory activity of sacran, plerixafor, enfuvirtide, and sacran with plerixafor or enfuvirtide. Combination indices of sacran with plerixafor (**B**) and sacran with enfuvirtide (**C**). The synergy between sacran and HIV-1 entry inhibitors was calculated using CompuSyn software based on the Chou–Talalay method [26,27]. Data represent the mean ± SD. Representative results from three independent experiments are shown.

**Figure 5 viruses-16-01501-f005:**
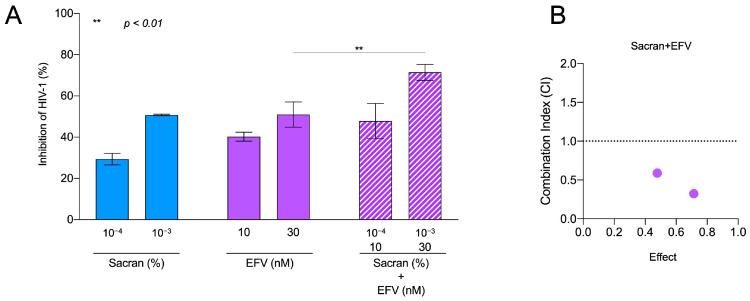
Synergistic HIV-1 inhibition of sacran and EFV. (**A**) HIV-1 inhibitory activity of sacran, EFV, and sacran with EFV. (**B**) Combination indices of sacran with EFV. The synergy between sacran and HIV-1 entry inhibitors was calculated using CompuSyn software based on the Chou–Talalay method [26,27]. Data represent the mean ± SD. Representative results from three independent experiments are shown.

**Table 1 viruses-16-01501-t001:** Summary of the combination index.

Sacran (%)	Plerixafor (nM)	Effect	Combination Index
10^−5^	10	0.178	0.72172
10^−4^	30	0.355	0.58724
10^−3^	50	0.623	0.32316
Sacran (%)	Enfuvirtide (nM)	Effect	Combination Index
10^−5^	30	0.443	0.55077
10^−4^	50	0.606	0.51666
10^−3^	100	0.781	0.51390
Sacran (%)	EFV (nM)	Effect	Combination Index
10^−4^	10	0.478	0.58724
10^−3^	30	0.714	0.32316

## Data Availability

All data supporting this study are included in the article. Further inquiries should be directed to the corresponding author.

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
