# Peer review of "Evaluating the Use of Sacran, a Polysaccharide Isolated from Aphanothece sacrum, as a Possible Microbicide for Preventing HIV-1 Infection"

_viruses, 2024, doi:10.3390/v16091501_

Round 1
Reviewer 1 Report
Comments and Suggestions for Authors
In this study, Matsuda et al. demonstrated the effects of sacran, a polysaccharide obtained from the cyanobacteria Aphanothece sacrum which has been used in skincare products and as dietary food, as an anti-HIV-1 agent in an in vitro setting. By using a sacran gel formulation, the authors showed its ability to inhibit HIV-1 infection, membrane fusion of infected cells in a fusion model system, and viral diffusion. Moreover, they also described a synergistic effect when used in combination with other well-described anti-HIV-1 drugs. The authors concluded that sacran gel could potentially be used as a microbicide and protect against HIV-1 transmission.
The data presented by the authors is clear and concise and support their conclusions. However, the following points need to be addressed by the authors before being suitable for publication:
1) More information about the sacran gel used in this study and its formulation should be included. The references provided in the materials section are not enough, as they use sacran in diverse settings (as viscous sacran solution, in heterogels with PVA or forming gel beads with several metal ions).
2) A major concern is the lack of a vehicle control (i.e. a control/placebo gel) in the experiments. It is needed to rule out the possibility that the observed effects are the result of the gel formulation, viscosity, etc. and not the sacran molecule itself.
3) The toxicity of the molecule should be addressed in the study, even if it is in a supplementary figure.
Other comments:
1) Since PM1-CCR5 cells were used for the infection assays, has a R5 tropic virus been tried? It would also be interesting to see the effect of sacran on PBMCs.
2) In the viral diffusion assays a higher sacran concentration than in previous experiments was used. Is sacran it not effective at 0.05-0.2%?
3) Likewise, sacran concentration in the synergism experiments is 50 to 5000 times lower than the infection assays. Is that because toxicity? Is synergism still observed at 0.05-0.2%?
4) It would be interesting for the authors to comment a bit more about their compound in the Discussion. Why do they think it has the effects it does? Is it different to other known microbicides?
Minor comments:
1) In Figure S1 legend, last sentence “(Materials and Methods :2.4. Flow cytometric detection of HIV-1-infected cells, Figure 1)” is not necessary as the figure is already referenced in the Materials and methods section itself.
2) The sentence from line 50 to 52 (“Sacran has potential as…”) should be rewritten. As written, it seems that article [11] has already described sacran as an anti-HIV-1 agent (which it has not).
3) Line 167, “(Figure 1B, C)” should be “(Figure 1B)”.
4) Line 170, “(Figure 1B, C)” should be “(Figure 1C)”.
5) Line 173, “Figure 1B and D)”, should be “(Figure 1D)”.
6) Mention the cell lines used in Results 3.2 section (MOLT-4 and Jurkats) for clarity. Also add that Jurkat 522F/Y is a fusion negative control.
7) Title of Results 3.3 section seems shifted. It should go below Figure 2.
8) There are two Figures 4. Synergism between Sacran and EFV should be Figure 5.
9) Figure 5A: FEV (nM) should be EFV (nM).
10) Sentence from line 261 to 264 (“Among young women…”) should be deleted. Avoid terms as in “PLWH are spreading it” as it puts the blame in PLWH. Moreover, “(HIV is) spreading due to criminal heterosexual and homosexual transmission” is a highly controversial and offensive statement, as it seems to put the blame of HIV pandemic on prostitution and homosexuality.
11) Line 311, “against dry sacran skin” should be “against dry skin”.
12) Line 311, reference [28] is duplicated.
13) Lines 310-312 are a bit misleading. Referenced article reported those effects against induced-allergic dermatitis in mice. In humans, only reduced transepidermal water loss in dry skin individuals was reported.
Author Response
Reviewer #1's comments:
In this study, Matsuda et al. demonstrated the effects of sacran, a polysaccharide obtained from the cyanobacteria Aphanothece sacrum which has been used in skincare products and as dietary food, as an anti-HIV-1 agent in an in vitro setting. By using a sacran gel formulation, the authors showed its ability to inhibit HIV-1 infection, membrane fusion of infected cells in a fusion model system, and viral diffusion. Moreover, they also described a synergistic effect when used in combination with other well-described anti-HIV-1 drugs. The authors concluded that sacran gel could potentially be used as a microbicide and protect against HIV-1 transmission.
The data presented by the authors is clear and concise and support their conclusions. However, the following points need to be addressed by the authors before being suitable for publication:
1) More information about the sacran gel used in this study and its formulation should be included. The references provided in the materials section are not enough, as they use sacran in diverse settings (as viscous sacran solution, in heterogels with PVA or forming gel beads with several metal ions).
【Reply】Thank you very much for taking the time to review this manuscript. We are happy to know that you judged that the significance of this research in this field is high. Also, we sincerely appreciate your constructive suggestions. We have taken your suggestion and added some additional references about the information of sacran gel in the Introduction section.
(Page 2, Introduction section) Page 2, Line70-
Sacran is considered high biocompatible because it is a polysaccharide derived from edible cyanobacteria, and is used in a variety of states including viscous solutions, heterogels, and gel beads. This gel-forming and film-forming ability is particularly unique, and heterogels conditioned with polyvinyl alcohol (PVA) have high degree of swelling and improve the water absorption of sacran (Okajima M et al. Biomacromolecules. 2010). Sacran also has excellent metal adsorption, more efficiently adsorbing heavy metal ions such as indium, rare earth metal, and lead ions to form gel beads (Okajima M et al. Langmuir. 2009).
In this study, we investigated the effect of a viscous sacran gel as an anti-HIV-1 agent using a transwell mimic of HIV-1 transmission in vaginal tissue.
2) A major concern is the lack of a vehicle control (i.e. a control/placebo gel) in the experiments. It is needed to rule out the possibility that the observed effects are the result of the gel formulation, viscosity, etc. and not the sacran molecule itself.
【Reply】 In this study, PBS was used for vehicle control. The reason is that sacran is not added as a drug to a gel-like substance such as hyaluronic acid, but the sacran molecule itself is in a gelatinized state. Therefore, we are confident that the effect shown in this study is the antiviral activity of the sacran molecule itself, including its physical properties.
3) The toxicity of the molecule should be addressed in the study, even if it is in a supplementary figure.
【Reply】 Thank you for your suggestion regarding the toxicity. We examined the toxicity of sacran gel in several cell lines (new Supplemental Figure 1) and described in the Result section. We also added the above assay methods in the Materials and Methods section. Toxicity studies using three cell lines demonstrated that sacran gel up to 0.2% had no effect on cell viabilities.
(Page 2, Materials and Methods section)
2.2. Cell lines and culture
TZM-bl, JurkatHXBc2 (4), Jurkat522F/Y, and BCBL-1 cells were procured from the National Institutes of Health AIDS Reagent Program (Bethesda, MD, USA). Molt-4 T cell line was obtained from RIKEN cell bank (Tsukuba, Japan). MT-4 T cell line was kindly provided by Dr. Kazuhiko Ide (Kumamoto University, Kumamoto, Japan). The human T cell line PM1-CCR5, stably expressing human CCR5 [15], was kindly gifted by Dr. Yosuke Maeda (Kumamoto University, Kumamoto, Japan). M213 human cholangiocarcinoma cell line was obtained from Japanese Collection of Research Bioresources (JCRB) Cell Bank (Osaka, Japan). Ihara human melanoma cell line was kindly provided by Dr. Keiichi Motoyama (Kumamoto University, Kumamoto, Japan). MT-4, MOLT-4, PM1-CCR5, and BCBL-1 cells were maintained in RPMI 1640 medium (Fujifilm Wako Pure Chemical, Osaka, Japan) supplemented with penicillin 100 U/mL, streptomycin 100 μg/mL, and 10% fetal bovine serum (Thermo Fisher Scientific, Waltham, MA, USA) and cultured in a 5% CO2 humidified incubator at 37 °C. Stably expressing HIV-1 env cell lines, JurkatHXBc2 (4), and Jurkat522F/Y were used for cell-cell fusion assay and cultured in RPMI 1640 supplemented with 10% FBS, 100 U/mL penicillin, 100 µg/mL streptomycin, 1 µg/mL doxycycline, 200 µg/mL G418, and 200 µg/mL hygromycin. JurkatHXBc2 (4) expresses functional gp120 and gp41 glycoproteins (Env), whereas Jurkat522F/Y has an F/Y substitution at position 522 of gp41, which prevents fusion [16]. HIV-1 env expression was induced by a Tet-Off system, whereby cells were washed with PBS and cultured in doxycycline-free medium for 3 d prior to fusion experiments[17, 18]. TZM-bl, M213, and Ihara cells were maintained in DMEM (Fujifilm Wako Pure Chemical) supplemented with penicillin 100 U/mL, streptomycin 100 μg/mL, and 10% fetal bovine serum.
(Page 3, Materials and Methods section)
2.5. Cytotoxicity assay
To examine cytotoxicity of sacran gel, several human cell lines, BCBL-1 (human B cell lymphoma), M213 (human cholangiocarcinoma), and Ihara (human melanoma) cells (5×105 cells/ml) were cultured in the presence or absence of sacral gel (final concentration 0-0.2%). After 24 h, a cytotoxicity assay (WST-8 assay) was performed using Cell Counting Kit-8 (Dojindo, Kumamoto, Japan) according to the manufacturer’s instructions. The number of viable cells after sacran gel treatment was measured and compared with untreated cells.
(Page 5, Results section)
Cytotoxicity of sacran gel was also examined using several human cell lines, BCBL-1 (human B cell lymphoma), M213 (human cholangiocarcinoma), and Ihara (human melanoma). Sacran gel was found to have no toxicity in these cells at concentrations up to 0.2% (see Figure S1).
Other comments:
- Since PM1-CCR5 cells were used for the infection assays, has a R5 tropic virus been tried? It would also be interesting to see the effect of sacran on PBMCs.
【Reply】Thank you for your comments. As reviewer mentioned, it is very critical to evaluate the activity of the R5-tropic virus, but in this study, we only tested HIV infection withX4-tropic virus (HIV-1NL4-3), as we used cells expressing the env derived from X4-tropic virus (HIV-1HXBc2) in the cell fusion assay. We expect sacran also blocks R5-tropic virus infection from the mechanism, and we will try the inhibition of both X4- and R5-tropic HIV infection against primary cells for the next issue.
- In the viral diffusion assays a higher sacran concentration than in previous experiments was used. Is sacran it not effective at 0.05-0.2%?
【Reply】 In the viral diffusion assay, a high efficacy was observed when high concentrations of sacran were used. We have added the results in the low concentration range (0.05-0.2%) and replaced it with the new Figure 3.
- Likewise, sacran concentration in the synergism experiments is 50 to 5000 times lower than the infection assays. Is that because toxicity? Is synergism still observed at 0.05-0.2%?
【Reply】The synergism experiments were conducted at lower concentrations of sacran than those used in the past, but this is not due to toxicity. The "Chou-Talalay method" is used for this evaluation, and since it is necessary to match the effects of the two drugs to be evaluated to the same level, the concentrations are matched to three levels of activity: weak, moderate, and high. 0.05-0.2 % concentration of sacran is sufficient for the evaluation of the effect of the two drugs. In the case of 0.05-0.2% sacran, the inhibitory effect of sacran on infection is too high and no synergistic effect is observed. In Figure 4, the concentrations of the drugs were matched so that sacran in the low 10-5 % to 10-3 % range had a 4 %, 11 %, and 40 % inhibitory effect on infection, while AMD3100 at 10 nM-50 nM and T-20 at 30 nM-100 nM had 17 %, 17 %, 43 %, and 28 %, 42 %, and 59 % inhibition, respectively. The concentrations of the drugs were matched to each other.
4) It would be interesting for the authors to comment a bit more about their compound in the Discussion. Why do they think it has the effects it does? Is it different to other known microbicides?
【Reply】 In previous microbicide development studies, it has been reported that microbicides containing tenofovir-containing gel (Karim QA et al. Science. 2010), an NRTI, and entry inhibitors such as CCR5 inhibitors are effective in inhibiting HIV infection through topical transmucosal application (Lederman M et al. Science. 2004; Veazey et al. Nature. 2005). We also examined the synergistic effects of efavirenz, an NRTI, and AMD3100 and T-20, entry inhibitors, and observed synergistic effects with sacran in all three drugs evaluated.
Minor comments:
1) In Figure S1 legend, last sentence “(Materials and Methods :2.4. Flow cytometric detection of HIV-1-infected cells, Figure 1)” is not necessary as the figure is already referenced in the Materials and methods section itself.
【Reply】 We deleted the last sentence in figure S1 legend “Materials and Methods :2.4. Flow cytometric detection of HIV-1-infected cells, Figure 1”.
2) The sentence from line 50 to 52 (“Sacran has potential as…”) should be rewritten. As written, it seems that article [11] has already described sacran as an anti-HIV-1 agent (which it has not).
【Reply】We changed the sentence from line 64 to 66 according to the reviewer's suggestion
3) Line 167, “(Figure 1B, C)” should be “(Figure 1B)”.
【Reply】We corrected it.
4) Line 170, “(Figure 1B, C)” should be “(Figure 1C)”.
【Reply】We corrected it.
5) Line 173, “Figure 1B and D)”, should be “(Figure 1D)”.
【Reply】We corrected it.
6) Mention the cell lines used in Results 3.2 section (MOLT-4 and Jurkats) for clarity. Also add that Jurkat 522F/Y is a fusion negative control.
【Reply】The cell lines used in Results section 3.2 (MOLT-4 and Jurkats) are clearly described, and we have added that Jurkat 522F/Y is a fusion negative control.
7) Title of Results 3.3 section seems shifted. It should go below Figure 2.
【Reply】We corrected it.
8) There are two Figures 4. Synergism between Sacran and EFV should be Figure 5.
【Reply】We changed figure number of the synergism between sacran and EFV to Figure 5.
9) Figure 5A: FEV (nM) should be EFV (nM).
【Reply】We corected it.
10) Sentence from line 261 to 264 (“Among young women…”) should be deleted. Avoid terms as in “PLWH are spreading it” as it puts the blame in PLWH. Moreover, “(HIV is) spreading due to criminal heterosexual and homosexual transmission” is a highly controversial and offensive statement, as it seems to put the blame of HIV pandemic on prostitution and homosexuality.
【Reply】We deleted this sentence.
11) Line 311, “against dry sacran skin” should be “against dry skin”.
【Reply】We corrected it.
12) Line 311, reference [28] is duplicated.
【Reply】We deleted it.
13) Lines 310-312 are a bit misleading. Referenced article reported those effects against induced-allergic dermatitis in mice. In humans, only reduced transepidermal water loss in dry skin individuals was reported.
【Reply】We changed the sentence from lines 337 to 339.
(Page 9, Lines 337-339 Discussion section) Furthermore, it has been reported to have against induced-allergic dermatitis in mice. In addition, sacran significantly reduced transepidermal water loss in individuals with dry skin has been reported in human [36].
Reviewer 2 Report
Comments and Suggestions for Authors
The article studies the antiviral activity of sacran, as well as the possibilities of its combined use with other anti-HIV-1 drugs. Sacran is a megamolecular polysaccharide extracted from cyanobacterium Aphanothece sacrum that exhibits numerous desirable characteristics for transdermic applications, such as safety as a biomaterial, a high moisture retention effect, the ability to form a film and hydrogel, and an anti-inflammatory effect. The developed approach, which involves the use of sacran hydrogel for protection against HIV-1 transmission, is very relevant and indicates the presence of an antiviral effect when using sacran and a synergistic effect of sacran and anti-HIV-1 drugs in vitro models. The usefulness of this study, which suggests the potential of sacran gel to protect against HIV-1 transmission, is beyond doubt. However, the effectiveness of sacran gel against HIV-1 infection in vivo requires further study.
From the point of view of the conducted research, the work, in general, makes a good impression and does not cause major complaints. However, questions related to design of this manuscript still arise.
1) The specification of the composition of sacran (lines 45-49) requires citation of sources.
2) Lines 52-53. The phrase "Sacran is a type of massively cultivated cyanobacteria..." is better formulated as "Sacran is produced..."
3) Line 311. The reference to the literature is indicated as [28][28].
4) The formatting of the references does not comply with MDPI Style.

Author Response
Reviewer #2's comments:
The article studies the antiviral activity of sacran, as well as the possibilities of its combined use with other anti-HIV-1 drugs. Sacran is a megamolecular polysaccharide extracted from cyanobacterium Aphanothece sacrum that exhibits numerous desirable characteristics for transdermic applications, such as safety as a biomaterial, a high moisture retention effect, the ability to form a film and hydrogel, and an anti-inflammatory effect. The developed approach, which involves the use of sacran hydrogel for protection against HIV-1 transmission, is very relevant and indicates the presence of an antiviral effect when using sacran and a synergistic effect of sacran and anti-HIV-1 drugs in vitro models. The usefulness of this study, which suggests the potential of sacran gel to protect against HIV-1 transmission, is beyond doubt. However, the effectiveness of sacran gel against HIV-1 infection in vivo requires further study.
From the point of view of the conducted research, the work, in general, makes a good impression and does not cause major complaints. However, questions related to design of this manuscript still arise.
【Reply】 Thank you for your review. We are happy to know that you judged that the significance of this research in this field is high. Also, we sincerely appreciate your constructive suggestions. In this study, we evaluated the anti-HIV efficacy of sacran in an in vitro model. Although the efficacy of sacran gel has not yet been demonstrated using a vaginal infection system in animal models such as immunodeficient mice and monkeys, we intend to develop a practical application of sacran gel in the future.
1) The specification of the composition of sacran (lines 45-49) requires citation of sources.
【Reply】We added the citation of sources (lines 59-61). Puluhulawa LE et al. Molecules. 26(11):3362. 2021. [14]
2) Lines 52-53. The phrase "Sacran is a type of massively cultivated cyanobacteria..." is better formulated as "Sacran is produced..."
【Reply】We changed the sentence from lines 66-68.
3) Line 311. The reference to the literature is indicated as [28][28].
【Reply】 We corrected it.
4) The formatting of the references does not comply with MDPI Style.
【Reply】 We adapted the format with MDPI style.
Reviewer 3 Report
Comments and Suggestions for Authors
Matsuda et al. present the results of a study on sacran, a polysaccharide isolated from the Aphanothece sacrum, as a potential vaginal microbicide for HIV prevention. The relevance of this topic is high. A series of experiments were conducted to demonstrate that sacran could inhibit HIV cell-to-cell fusion and transmission. However, the manuscript contains serious issues that must be addressed.
Major comments:
1. The current title is complex and should be shortened. The word “potent” should be omitted to ensure the title remains neutral one. For instance, “Sacran, a polysaccharide isolated from Aphanothece sacrum, as a possible microbicide for the prevention of HIV infections”.
2. The “Introduction” section requires significant rewriting. It should address previous pre-exposure prophylaxis (PrEP) approaches in greater depth than it currently does (lines 34-41), reflecting the failures in this field. In addition, it is curious to me why the introduction specifically focuses on women? (lines 32-37), as women are not the only ones affected by HIV. A better balance in these sentences is needed for clarity. Moreover, important reviews in this field are missing, for instance, 10.1371/journal.pmed.0020142, 10.1111/tmi.13401.
3. Reference 11 in the Introduction does not mention sacran at all. Additionally, the sentence in lines 50-52 “Sacran has potential as an anti-HIV-1 agent because a previous study demonstrated that the advantages of its anti-HIV-1 activity include its high molecular weight and high degree of sulfation” is confusing and should be rephrased. High molecular weight and a high degree of sulfation are not advantages of the molecule for biological activity. As a result, the rationale for the studying sacran as a microbicide for HIV prevention is unclear.
4. The “Materials and Methods” section should specify the source of the compounds used as reference drugs – AMD3100, T-20, and efavirenz. More importantly, please use international nonproprietary names for these drugs – plerixafor for AMD3100 and enfuvirtide for T-20 – to avoid misunderstandings.
5. Line 58: “… in vaginal tissue.”. However, the “Cell lines and culture” subsection do not include any vaginal cell lines. It would be beneficial to conduct a transwell assay using vaginal cells, for instance, PCS-480-010. Due to these methodological issues in the in vitro study, the potential application of sacran as a vaginal microbicide is not well supported.
6. What is the main reason for conducting a synergy assay of sacran with HIV-1 entry inhibitors? Sacran is a possible option for pre-expositie profylaxisis (PrEP) that is used topically before HIV exposure, while HIV-1 entry inhibitors (plerixafor and enfuvirtide) and HIV-1 reverse transcriptase inhibitors (efavirenz) are components for highly active antiretroviral therapy (HAART) that are used subcutaneously or orally after HIV exposure. In animal and human studies, it would be very difficult to assess the effects of this combination.
7. In my opinion, Figure S1 is important and should be moved to the main text, combined with Figure 1.
8. Figures 1 and 2: Appropriate HIV drugs as reference controls are missing in these experiments. The lack of proper reference compounds is a methodological error of the study.
9. Figure 3: Appropriate control, for instance, tenofovir gel, is missing in a transwell assay.
10. Lines 308-312: Unfortunately, these findings do not indicate the safety of sacran. The safety of sacran (and any compound) must be established through specific (acute and chronic) toxicity studies in animals (rodents and non-rodents) as well as in Phase 1 clinical trials. Please delete the sentence “These findings indicate the safety of sacran” to avoid misunderstandings.
Minor comments:
1. The abstract, as well as the whole text, should avoid abbreviations without their prior description.
2. Line 29 “…several products to prevent…” – What products are being referred to here? I suggest replacing the word “products” with “HIV testing technologies” for clarity, as the cited reference only discusses HIV testing and universal precautions.
3. Lines 34-37, line 55: the proper references are needed.
4. Line 119: The name of the cell culture used is missing.
5. Figure 1, Figure 2, Figure 3: It is unclear what groups were compared. Please clarify this information.
6. Line 201 and 219 “Transwell plates were incubated in a 5% 219 CO2 incubator at 37 ◦C for 24 h”: this sentence should be in the protocol, not in the “Results” section.
7. Figure 4: FEV should be corrected to EFV.
8. Table 1: please ensure that sacran and the other drugs are indicated in the same metric system (nM).
9. The sentence “The development of tools to prevent viral infections has previously shown that TFV-based gels, such as PrEP, can decrease HIV-1 infection acquisition.” should be rewritten for clarity.
10. Lines 287: Please omit the word “novel” when referring to sacran, as it was first mentioned in 2009.
Comments on the Quality of English LanguageSee my Comments and Suggestions for Authors
Author Response
Reviewer #3's comments:
Matsuda et al. present the results of a study on sacran, a polysaccharide isolated from the Aphanothece sacrum, as a potential vaginal microbicide for HIV prevention. The relevance of this topic is high. A series of experiments were conducted to demonstrate that sacran could inhibit HIV cell-to-cell fusion and transmission. However, the manuscript contains serious issues that must be addressed.
【Reply】 Thank you very much for taking the time to review this manuscript. We are happy to know that you judged that the significance of this research in this field is high. Also, we sincerely appreciate your constructive suggestions. We have made revisions based on your suggestions.
Major comments:
- The current title is complex and should be shortened. The word “potent” should be omitted to ensure the title remains neutral one. For instance, “Sacran, a polysaccharide isolated from Aphanothece sacrum, as a possible microbicide for the prevention of HIV infections”.
【Reply】 Thank you very much for your comment. We have shortened and changed the title according to your suggestion.
- The “Introduction” section requires significant rewriting. It should address previous pre-exposure prophylaxis (PrEP) approaches in greater depth than it currently does (lines 34-41), reflecting the failures in this field. In addition, it is curious to me why the introduction specifically focuses on women? (lines 32-37), as women are not the only ones affected by HIV. A better balance in these sentences is needed for clarity. Moreover, important reviews in this field are missing, for instance, 10.1371/journal.pmed.0020142, 10.1111/tmi.13401.
【Reply】 We have changed the text based on your suggestion. According to a 2017 study, the number of newly infected adolescent girls and young women (AGYW) aged 15-24 is very high, about 1,000 per day. In sub-Saharan Africa in particular, AGYW are twice as likely to be infected with HIV as men, and 75% of new infections among adolescents aged 15-19 are reported to be girls (Musekiwa A et al. Trop Med Int Health. 2020). Male condoms are the most widely publicized HIV prevention measure, but this relies on the way men manage it, which may lead to limiting the ability of more vulnerable women to manage their own HIV prevention.
Pre-exposure pro-phylaxis (PrEP) is a very effective means of HIV prevention, but it requires strict adherence to daily medication regimens, which has been shown to fail in some cases. The emergence of drug-resistant viruses with PrEP has also been reported (Parikh UM et al. Curr Opin HIV AIDS. 2022), but adherence problems have been pointed out as a possible cause, and it can be difficult to manage a fixed schedule of oral dosing.
Microbicide is a potential female-led biomedical intervention for HIV prevention when the female side cannot negotiate condom use. On the other hand, there remains a high prevalence among men who have sex with men (MSM), and frequent anal intercourse is a risk factor. Therefore, microbicide may contribute to HIV infection risk reduction not only for women but also for men.
(Page 1, Lines 34-43, Introduction section)
Most human HIV-1 infections are sexually transmitted through infected semen and vaginal or cervical secretions containing infected lymphocytes [2]. According to a 2017 study, the number of newly infected adolescent girls and young women (AGYW) aged 15-24 is very high, about 1,000 per day. In sub-Saharan Africa in particular, AGYW are twice as likely to be infected with HIV as men, and 75% of new infections among adolescents aged 15-19 are reported to be girls [3]. Male condoms are the most widely publicized HIV prevention measure, but this relies on the way men manage it, which may lead to limiting the ability of more vulnerable women to manage their own HIV prevention. Pre-exposure pro-phylaxis (PrEP) is a very effective means of HIV prevention, but it requires strict adherence to daily medication regimens, which has been shown to fail in some cases. The emergence of drug-resistant viruses with PrEP has also been reported [4], but adherence problems have been pointed out as a possible cause, and it can be difficult to manage a fixed schedule of oral dosing.
Lines 44-51
Microbicides can be applied to the vagina and rectum to reduce the transmission of sexually transmitted infections (STIs), including HIV-1. Vaginal microbicides are a potential means of protecting women against HIV-1 infection during sexual transmission [5]. Thus, microbicide is a potential female-led biomedical intervention for HIV prevention when the female side cannot negotiate condom use. On the other hand, there remains a high prevalence among men who have sex with men (MSM), and frequent anal intercourse is a risk factor. Therefore, microbicide may contribute to HIV infection risk reduction not only for women but also for men.
- Reference 11 in the Introduction does not mention sacran at all. Additionally, the sentence in lines 50-52 “Sacran has potential as an anti-HIV-1 agent because a previous study demonstrated that the advantages of its anti-HIV-1 activity include its high molecular weight and high degree of sulfation” is confusing and should be rephrased. High molecular weight and a high degree of sulfation are not advantages of the molecule for biological activity. As a result, the rationale for the studying sacran as a microbicide for HIV prevention is unclear.
【Reply】 Thank you very much. We rephrased the sentenceas follows (Lines 64-66).
(Page 2, Lines 64-66, Introduction section) Previous studies have shown that sacran has a high molecular weight and high degree of sulfation, which is an advantage for its potential as an anti-HIV-1 agent [15].
- The “Materials and Methods” section should specify the source of the compounds used as reference drugs – AMD3100, T-20, and efavirenz. More importantly, please use international nonproprietary names for these drugs – plerixafor for AMD3100 and enfuvirtide for T-20 – to avoid misunderstandings.
【Reply】 According to the reviewr’s suggestion, we have clearly stated the source of the compounds used as reference drugs, AMD3100, T-20, and efavirenz, and changed the names of these drugs to their international non-proprietary names.
- Line 58: “… in vaginal tissue.”. However, the “Cell lines and culture” subsection do not include any vaginal cell lines. It would be beneficial to conduct a transwell assay using vaginal cells, for instance, PCS-480-010. Due to these methodological issues in the in vitro study, the potential application of sacran as a vaginal microbicide is not well supported.
【Reply】 Thank you for your comment. As you mentioned, we did not perform the assay using vaginal tissue in this experiment, so we deleted the phrase "in vaginal tissue" and changed it to "macrophage/dendritic cells and T lymphocytes in mucosal tissue, since vaginal cells are not direct target of HIV.
(Page 2, Lines77-80, Introduction section) In this study, we investigated the effect of a viscous sacran solution as an anti-HIV-1 agent using a transwell that mimic HIV-1 transmission by macrophages/dendritic cells and T lymphocytes in mucosal tissue.
- What is the main reason for conducting a synergy assay of sacran with HIV-1 entry inhibitors? Sacran is a possible option for pre-expositie profylaxisis (PrEP) that is used topically before HIV exposure, while HIV-1 entry inhibitors (plerixafor and enfuvirtide) and HIV-1 reverse transcriptase inhibitors (efavirenz) are components for highly active antiretroviral therapy (HAART) that are used subcutaneously or orally after HIV exposure. In animal and human studies, it would be very difficult to assess the effects of this combination.
【Reply】 Several HIV entry inhibitor microbicides have been formulated. In particular, it has been reported that a gel formulation of maraviroc, a CCR5 inhibitor, can inhibit HIV infection via the vaginal route in humanized mice (Neff CP et al. PLoS one. 2011) and rhesus monkeys (Veazey RS et al. J Infect Dis. 2010). Reverse transcriptase inhibitor-based microbicides include a gel formulation using TDF, an NRTI (Karim QA et al. Science. 2010), and a gel based on dapivirine, an NNRTI, which has been demonstrated in preclinical studies to prevent HIV infection in humanized mice (Fabio SD et al. AIDS. 2003). In addition, a vaginal ring containing dapivirine has shown efficacy in human clinical trials (Baeten JM et al. N Engl J Med. 2016). In this study, we conducted a synergistic study of sacran with HIV entry inhibitor and NNRTI and found its usefulness as a candidate for microbicide.
- In my opinion, Figure S1 is important and should be moved to the main text, combined with Figure 1.
【Reply】 Thank you for your comment. As the reviewer suggestion, we have moved Figure S1 to the main text as Figure 1A.
- Figures 1 and 2: Appropriate HIV drugs as reference controls are missing in these experiments. The lack of proper reference compounds is a methodological error of the study.
【Reply】 In this study, PBS was used as the reference control. This is because sacran is not added as a drug to a gel-like substance such as hyaluronic acid, but the sacran molecule itself is in a gel state. Therefore, it can be said that the effect shown in this study is the antiviral action of the sacran molecule itself, including its physical properties.
- Figure 3: Appropriate control, for instance, tenofovir gel, is missing in a transwell assay.
【Reply】 We agree that tenofovir gel is the best control, but it was difficult to obtain tenofovir gel in Japan (It is not approved in Japan. Most of the HIV-infected individuals are MSM in Japan). In the transwell assay in Figure 3, we used without sacran gel (virus solution only) as a control.
- Lines 308-312: Unfortunately, these findings do not indicate the safety of sacran. The safety of sacran (and any compound) must be established through specific (acute and chronic) toxicity studies in animals (rodents and non-rodents) as well as in Phase 1 clinical trials. Please delete the sentence “These findings indicate the safety of sacran” to avoid misunderstandings.
【Reply】 According to the reviewer’s comment, we deleted the sentence “These findings indicate the safety of sacran”.
Minor comments:
- The abstract, as well as the whole text, should avoid abbreviations without their prior description.
【Reply】 We have spelled out abbreviations appropriately in the abstract.
combination antiretroviral therapy (cART) (Line 12), human immunedeficiency virus type-1 (HIV-1) (Line 13), Acquired immune deficiency syndrome (AIDS) (Line 14)
- Line 29 “…several products to prevent…” – What products are being referred to here? I suggest replacing the word “products” with “HIV testing technologies” for clarity, as the cited reference only discusses HIV testing and universal precautions.
【Reply】 Thank you for your comment. We have replaced the word “products” with “HIV-1 testing technologies” as you indicated (Line 30).
- Lines 34-37, line 55: the proper references are needed.
【Reply】 We have added appropriate references to each.
Line 45-46: Cutler B et al. Lancet Infect Dis. 8(11):685-97. 2008. [5]
new Line 61: Puluhulawa LE et al. Molecules. 26(11):3362. 2021.[14]
- Line 119: The name of the cell culture used is missing.
We have corrected the sentence as follows:
(Page 4, Line 167: Materials and Methods section) The bottom compartment of each Transwell contained 600 µL of10%FCS/DMEM.
- Figure 1, Figure 2, Figure 3: It is unclear what groups were compared. Please clarify this information.
【Reply】 According to the reviewer’s comment, we have added bars to clarify the groups we compared in Figure1CDE (Line208), Figure 2B (Line 231), and Figure 3BC (Line 246).
- Line 201 and 219 “Transwell plates were incubated in a 5% CO2 incubator at 37 ◦C for 24 h”: this sentence should be in the protocol, not in the “Results” section.
【Reply】Thank you for your comment, the same sentence is in the “Materials and Methods” section, and We have removed it from the “Results” section.
- Figure 4: FEV should be corrected to EFV.
【Reply】 We corrected it.
- Table 1: please ensure that sacran and the other drugs are indicated in the same metric system (nM).
【Reply】 Thank you for your comment. The sacran concentrations listed in Table 1 were changed to the same “metric system” as displayed in Figures 4 and 5.
- The sentence “The development of tools to prevent viral infections has previously shown that TFV-based gels, such as PrEP, can decrease HIV-1 infection acquisition.” should be rewritten for clarity.
【Reply】 Thank you for your comment. We rewrote the sentence.
(Page 8, Line 290-294; Discussion section) To the present, the development of tools to prevent viral transmission has previously shown that TFV-based gel formulations currently prescribed in PrEP reduce the chance of HIV-1 transmission; the CAPRISA 004 trial was the first large trial to demonstrate the efficacy of 1% TFV gel, which reduced 39% of heterosexual HIV-1 infections [7].
- Lines 287: Please omit the word “novel” when referring to sacran, as it was first mentioned in 2009.
【Reply】Thank you for your comment. We deleted the word “novel” in the lines 314.
Reviewer 4 Report
Comments and Suggestions for Authors
This manuscript reported the anti-HIV activity of a megamolecular polysaccharide called Sacran. The experiment was delicately designed and carefully implemented. The results were clearly explained and suggest a hope of future use. However, I have a bit of concerns.
1. The concentration of Sacran was different in different experiment. Sacran was 0.05%, 0.1%, 0.2% in inhibition of HIV infection experiment (Figure 1) and inhibition of HIV envelope-dependent cell fusion (Figure 2); but 0.5 and 1% for inhibition of diffusion and capture (Figure 3); and became 0.00001%, 0.0001%, 0.001% in the synergistic inhibition with entry inhibitor or EFV. I wish the authors could give some explanation the reason for this difference.
2. The synergistic effect of sacran and AMD3100 or T20 (Figure 4), are the P values between the inhibition of AMD3100 and AMD3100+Saran, T20 and T20+Sacran? Table 1, what is the meaning of Effect? How to read this information, the bigger the better? I wish the authors could give some description and explanation.
3. Line 141-142, the HIV-1 solution collected from the bottom chamber. Here, the bottom chamber was not clear because the 2.6 transwell experiment was for figure 3 results but not for Figure 4 and/or Figure 5. So the authors may need to add some more information about the experiment of synergistic effects.
Other minor issues:
1. Sacran+EFV figure should be Figure 5.
2. Figure 1 and legend, there is no **, *** here.
I think BCD result was from experiment (a) (b) (c). If so, I wish the authors could make it much clear.
(B)(D) Intracellular p24 (HIV Gag) levels in sacran-treated PM1-CCR5 cells. I think this is the percentage of p24 (HIV-infected) positive cells here. If so, I wish the authors could make it clear. Otherwise, p24 levels sounds like the fluorescence intensity here.
3. Line 205: (Figure 3B) may not be needed here.
4. Figure 4: There is no *** on any bars, so it could be deleted.
5. Line 311, pleas delete one [28].
6. Figure S1: Figure S1A, Figure S1B, Figure S1C. Figure S1 could be deleted.
Author Response
Reviewer #4's comments:
This manuscript reported the anti-HIV activity of a megamolecular polysaccharide called Sacran. The experiment was delicately designed and carefully implemented. The results were clearly explained and suggest a hope of future use. However, I have a bit of concerns.
【Reply】Thank you very much for taking the time to review this manuscript. We are happy to know that you judged that the significance of this research in this field is high. Also, we sincerely appreciate your constructive suggestions.
- The concentration of Sacran was different in different experiment. Sacran was 0.05%, 0.1%, 0.2% in inhibition of HIV infection experiment (Figure 1) and inhibition of HIV envelope-dependent cell fusion (Figure 2); but 0.5 and 1% for inhibition of diffusion and capture (Figure 3); and became 0.00001%, 0.0001%, 0.001% in the synergistic inhibition with entry inhibitor or EFV. I wish the authors could give some explanation the reason for this difference.
【Reply】 Regarding the different sacran concentrations in each experiment, a significant effect was observed in the viral diffusion assay when high concentrations of sacran were used. We have added the results of the low concentration range (0.05-0.2%) and replaced it with the new Figure 3. The synergism experiment was conducted at lower concentrations of sacran than those used in the past. The “Chou-Talalay method” is used for this evaluation. Since it is necessary to match the effects of the two drugs to be evaluated to the same degree, we have experimented by matching the concentrations so that there are three levels of activity: weak, medium, and high. 0.05-0.2 % concentration of sacran is sufficient to achieve the same level of activity. In Figure 4, the drug concentrations were adjusted to achieve 4 %, 11 %, and 40 % inhibitory effect of sacran in the low concentration range of 10-5 % to 10-3 %, 17 %, 17 %, and 43 % inhibitory effect of AMD3100 at 10 nM to 50 nM, and 28 %, 42 %, and 59 % inhibitory effect of T-20 at 30 nM to 100 nM.
- The synergistic effect of sacran and AMD3100 or T20 (Figure 4), are the P values between the inhibition of AMD3100 and AMD3100+Saran, T20 and T20+Sacran? Table 1, what is the meaning of Effect? How to read this information, the bigger the better? I wish the authors could give some description and explanation.
【Reply】 As you indicated, we will replace Figure 4 as a new Figure 4 with a clearer comparison of the significant differences. Similarly, we will replace Figure 5. We have compared the results of the drug alone with those of the drug + sacran combination. The meaning of “Effect” in Table 1 is calculated based on the analysis by CompuSyn software. The closer the value is to 1.0 (the larger the value), the higher the activity. In contrast, the Combination Index (CI) is closer to 0 (smaller), indicating a higher synergistic effect. Therefore, the higher the Effect and the lower the CI, the better.
- Line 141-142, the HIV-1 solution collected from the bottom chamber. Here, the bottom chamber was not clear because the 2.6 transwell experiment was for figure 3 results but not for Figure 4 and/or Figure 5. So the authors may need to add some more information about the experiment of synergistic effects.
【Reply】 As the reviewer mentioned, the 2.6 (changed to 2.7) transwell experiment is the result of Figure 3. The bottom chamber that is referred to in Figure 4 and/or Figure 5 is when the experiment was performed in “Materials and Methods” Section 2.7: Transwell experiments for TZM-bl assay (changed to 2.8). We have changed some text in the Materials and Methods section about this.
(Page 4 Line 174, Materials and Methods section)The HIV-1 solution collected from the bottom chambers (as shown in Materials and Methods section 2.8 above) was used to infect the cells.
Other minor issues:
- Sacran+EFV figure should be Figure 5.
【Reply】 We changed figure number of the synergism between sacran and EFV to Figure 5.
- Figure 1 and legend, there is no **, *** here.
I think BCD result was from experiment (a) (b) (c). If so, I wish the authors could make it much clear.
(B)(D) Intracellular p24 (HIV Gag) levels in sacran-treated PM1-CCR5 cells. I think this is the percentage of p24 (HIV-infected) positive cells here. If so, I wish the authors could make it clear. Otherwise, p24 levels sounds like the fluorescence intensity here.
【Reply】 The p-values of significant differences and the experimental outline have been corrected for clarity (see new Figure 1). (B) to (D) show the percentage of p24 (HIV-infected) positive cells (the Y-axis was missing).
- Line 205: (Figure 3B) may not be needed here.
【Reply】 We deleted it.
- Figure 4: There is no *** on any bars, so it could be deleted.
【Reply】 We replaced Figure 4 as a new Figure 4 with a clearer comparison of the significant differences.
- Line 311, please delete one [28].
【Reply】 We deleted it.
- Figure S1: Figure S1A, Figure S1B, Figure S1C. Figure S1 could be deleted.
【Reply】 As the reviewer indicated, we have deleted Figure S1, and move to it as Main Figure 1A.
Round 2
Reviewer 1 Report
Comments and Suggestions for Authors
The authors answered all the questions and considered all the comments. However, there are two issues that need to be further addressed. I apologize if I did not convey the message properly on the first review.
1. Regarding the “vehicle control” comment: Even though there is a control without sacran (PBS as clarified), PBS does not have the gel-forming ability of sacran. This may be an issue, specifically the viral diffusion assay, as particle diffusion is slower in gels or viscous liquids than in water. In your data, the viral diffusion was highly inhibited only at 0.5-1% of sacran. Given the gel-forming ability of sacran, I wonder whether higher %sacran results in higher viscosity, and whether that viscosity is the responsible for the anti-diffusion properties. Therefore, a control of viscosity equal to that of the sacran conditions is needed to rule out that possibility. If such a control is not possible, the viscosity issue should at least be noted and discussed somewhere in the paper.
2. Regarding the “toxicity” comment: I thank you for providing the toxicity data on three cell models. However, I do not understand why the cell lines tested (BCBL-1, M213 and Ihara cells) are not the ones used on the manuscript (PM1-CCR5, MOLT-4, Jurkat and TZM-bl cells). To show that sacran has antiviral activity without toxicity, the conditions for the toxicity assays (cell lines, sacran concentration and incubation time) should be the same as in the anti-HIV assays. Moreover, given that the anti-diffusion activity is observed at 0.5-1% sacran, those concentrations should also be tested for toxicity. May you please provide toxicity data on those cell lines, and at 0.05-1% of sacran?
After the authors address the previous comments, I think the manuscript would be ready for publication.
Author Response
The authors answered all the questions and considered all the comments. However, there are two issues that need to be further addressed. I apologize if I did not convey the message properly on the first review.
Comments 1: Regarding the “vehicle control” comment: Even though there is a control without sacran (PBS as clarified), PBS does not have the gel-forming ability of sacran. This may be an issue, specifically the viral diffusion assay, as particle diffusion is slower in gels or viscous liquids than in water. In your data, the viral diffusion was highly inhibited only at 0.5-1% of sacran. Given the gel-forming ability of sacran, I wonder whether higher %sacran results in higher viscosity, and whether that viscosity is the responsible for the anti-diffusion properties. Therefore, a control of viscosity equal to that of the sacran conditions is needed to rule out that possibility. If such a control is not possible, the viscosity issue should at least be noted and discussed somewhere in the paper.
Response 1: Thank you for your comments. We also appreciate your assessment of the improvements we have made in Revise. As for vehicle control, as we mentioned before, we only conducted experiments using PBS. We agree with the reviewer that we need a control with the same viscosity as the sacran conditions. We have discussed this issue in the text.
(Page 6, Results section, Line 236 to 260) A marked inhibition of viral diffusion has been observed at high concentrations (0.5-1%). This may be related to the viscosity of sacran and may involve the physical properties of sacran, which is a macromolecule.
Comments2: Regarding the “toxicity” comment: I thank you for providing the toxicity data on three cell models. However, I do not understand why the cell lines tested (BCBL-1, M213 and Ihara cells) are not the ones used on the manuscript (PM1-CCR5, MOLT-4, Jurkat and TZM-bl cells). To show that sacran has antiviral activity without toxicity, the conditions for the toxicity assays (cell lines, sacran concentration and incubation time) should be the same as in the anti-HIV assays. Moreover, given that the anti-diffusion activity is observed at 0.5-1% sacran, those concentrations should also be tested for toxicity. May you please provide toxicity data on those cell lines, and at 0.05-1% of sacran?
Response 2: Thank you also for your comments on toxicity. We performed the cytotoxicity assay using the cell lines used in the manuscript (PM1-CCR5, MOLT-4, Jurkat) under the same conditions as the anti-HIV assay (sacran concentration, incubation time). In particular, the concentration range with anti-diffusion activity (0.5-1%) was also performed, indicating that sacran has antiviral activity without toxicity.
(Page 3, Materials and Methods section)
2.5. Cytotoxicity assay
To examine cytotoxicity of sacran gel, several lineages of human T cell lines, PM1-CCR5, MOLT4, and Jurkat cells (5×105 cells/ml) were cultured in the presence or absence of sacral gel (final concentration 0-1%). After 24 h, a cytotoxicity assay (WST-8 assay) was performed using Cell Counting Kit-8 (Dojindo, Kumamoto, Japan) according to the manufacturer’s instructions. The number of viable cells after sacran gel treatment was measured and compared with untreated cells.
(Page 5, Results section, Lines196-) Cytotoxicity of sacran gel was also examined using several human T cell lines, PM1-CCR5, MOLT4, and Jurkat cells. Sacran gel was found to have no toxicity in these cells at concentrations up to 1% (see Figure S1).
Commnets3: After the authors address the previous comments, I think the manuscript would be ready for publication.
Response3: Thank you very much for your valuable comments. Our manuscript is now getting clear.
Reviewer 3 Report
Comments and Suggestions for Authors
In version 2, Matsuda et al. addressed almost all the comments. However, some major comments were not addressed in the revised manuscript.
1. Lines 44-45 “Accorgint and rectum to reduce the transmission of sexually transmitted infections 44 (STIs), including HIV-1”. This sentence sounds awkwardly, please rewrite it.
2. Lines 64-66 “Previous studies have shown that sacran has a high 64 molecular weight and high degree of sulfation, which is an advantage for its potential as 65 an anti-HIV-1 agent [15]”. This important comment, which relates to the work as a whole, remains unanswered. High molecular weight and a high degree of sulfation are NOT advantages of the molecule for biological activity. This raises the question of the rationale for studying sacran as a microbicide for HIV prevention.
3. The answer to questions 8 and 9 is unclear. All studies of potential new drugs must include appropriate control groups (commercially available drugs). This is a serious limitation that should be acknowledged.
Comments on the Quality of English LanguageSee comment 1
Author Response
Comments 1: Lines 44-45 “Accorgint and rectum to reduce the transmission of sexually transmitted infections 44 (STIs), including HIV-1”. This sentence sounds awkwardly, please rewrite it.
Response 1: Thank you for your comment. We changed the sentence according to the instructions.
(Page 1, Introduction section) Microbicides can be applied to the vagina and rectum to reduce the transmission of sexually transmitted infections (STIs), including HIV-1 [5].
Comments 2: Lines 64-66 “Previous studies have shown that sacran has a high 64 molecular weight and high degree of sulfation, which is an advantage for its potential as 65 an anti-HIV-1 agent [15]”. This important comment, which relates to the work as a whole, remains unanswered. High molecular weight and a high degree of sulfation are NOT advantages of the molecule for biological activity. This raises the question of the rationale for studying sacran as a microbicide for HIV prevention.
Response 1: Thank you for your comment. There may be some lack of evidence regarding this sentence. It has been removed in accordance with the reviewer's comments.
Comments2: The answer to questions 8 and 9 is unclear. All studies of potential new drugs must include appropriate control groups (commercially available drugs). This is a serious limitation that should be acknowledged.
Response 3 Thank you for your comment. As the reviewer mentioned, we agree that tenofovir gel is the best control. However, as we mentioned previously, this was difficult to obtain and we were unable to perform the experiment again.